# Impedance Coupled Voltage Boosting Circuit for Polyvinylidene Fluoride Based Energy Harvester

**DOI:** 10.3390/s23010137

**Published:** 2022-12-23

**Authors:** Kibae Lee, Yoonsang Jeong, Chong Hyun Lee, Jongkil Lee, Hee-Seon Seo, Yohan Cho

**Affiliations:** 1Department of Ocean System Engineering, Jeju National University, Jeju 63243, Republic of Korea; 2Department of Mechanical Engineering Education, Andong National University, Andong 36729, Republic of Korea; 3Agency for Defense Development, Daejeon 34186, Republic of Korea

**Keywords:** energy harvesting, polyvinylidene fluoride, impedance coupling, voltage boosting

## Abstract

Polyvinylidene fluoride (PVDF) is an emerging method for energy harvesting by fluid motion with superior flexibility. However, the PVDF energy harvester, which has a high internal impedance and generates a low voltage, has a large power transmission loss. To overcome this problem, we propose an impedance-coupled voltage-boosting circuit (IC-VBC) that reduces the impedance of the PVDF energy harvester and boosts the voltage. SPICE simulation results show that IC-VBC reduces the impedance of the PVDF energy harvester from 4.3 MΩ to 320 kΩ and increases the output voltage by 2.52 times. We successfully charged lithium-ion batteries using the PVDF energy harvester and IC-VBC with low-speed wind power generation.

## 1. Introduction

Recently, the demand for renewable energy and the internet of things (IoT) has increased and this has urged the development of independent energy production technology for wearable devices. For energy harvesting, research using piezoelectricity, thermoelectricity, and triboelectricity is being conducted [1,2,3,4,5,6]. Especially, Piezoelectric elements are emerging materials for miniaturized energy generation with small size and high energy efficiency [7,8,9]. Polyvinylidene fluoride (PVDF) has been studied because it can generate low-speed wind power with a small size compared to conventional wind power generators [10,11,12,13,14,15,16]. However, the PVDF, which has a high internal impedance with thin thickness and generates a low voltage, causes a large leakage of power [17].

A rectifier circuit using a bridge diode is mainly used to deliver AC power generated by piezoelectric elements [18,19]. However, the bridge diode rectifier circuit causes power loss due to the forward voltage drop of diodes.

To overcome this power loss, a voltage doubler rectifier circuit was used [20,21]. The voltage doubler rectifier circuit using two diodes and capacitors can double the voltage produced by piezoelectric elements. However, it can cause power loss in energy harvesting using PVDF because it increases the intrinsic impedance of the harvesting circuit.

A voltage multiplier circuit and voltage-controlled switch were used for efficient power transfer of the PVDF energy harvester [22]. This method has practical limitations in that power cannot be continuously transferred and additional power is consumed for switching operation. Impedance coupling methods using a switch have been proposed to transfer power from an energy source with high impedance [23,24,25]. However, these methods have limited use for high-voltage sources and consume additional power for switching operations. Liang et al. reduced the impedance of the energy source without additional power consumption using capacitors and MOSFET transistors, but it can be used limitedly in harvesters that generate high voltage [26]. A circuit using a transformer for impedance matching was proposed by Zhu et al. [27]. This circuit can only be used for high input frequencies of several kHz and usually uses a large custom-made transformer of high inductance.

Cockcroft-Walton cascade voltage doubler circuit and Karthaus-Fischer cascade voltage doubler circuit were used for low-power energy harvesting circuits adopting Piezoelectric transducers [28]. Duque et al. proposed a voltage elevator circuit using two capacitors and an active rectifier circuit reducing forward voltage drop [29]. A charge pump rectifier circuit using many diodes and capacitors is used for low-power energy harvesting using photovoltaic cells, thermoelectric generators, etc. [30,31]. These methods increase the voltage efficiently by using multiple capacitors, but increase the impedance of the harvester and cause transmission loss of power.

To overcome this problem, we propose an impedance-coupled voltage-boosting circuit (IC-VBC) that reduces the impedance of the PVDF energy harvester and boosts the voltage. The proposed IC-VBC consists of a voltage-boosting stage that increases voltage and a storing stage that transfers power by coupling impedance. The voltage-boosting stage stores power in multiple capacitors connected in parallel and boosts the voltage by switching to a series connection. The storing stage implements a switch without power consumption by using MOSFET transistors and delivers power by matching the impedance to the electrical load. The proposed IC-VBC reduced the internal impedance of the PVDF energy harvester and increased the voltage to successfully generate low-speed wind power.

Our paper is organized as follows: Section 2 describes the IC-VBC. In Section 3, we design IC-VBC and describe the SPICE simulation results. In Section 4, the experimental results with low-speed wind generation are described and lithium-ion battery charging is demonstrated. Conclusions are presented in Section 5.

## 2. IC-VBC

Figure 1 shows the working principle of the proposed IC-VBC. The differential signal $V_{s}$ generated by the PVDF energy harvester is stored in K capacitors $C_{k}^{P}$ and $C_{k}^{N}$ of the equal capacitance connected in parallel according to the current direction. Based on the clockwise current flow, $C_{k}^{P}$ are charged while $V_{s}$ is a positive voltage and $C_{k}^{N}$ are charged while $V_{s}$ is a negative voltage. The total capacitances of $C_{parallel}^{P}$ and $C_{parallel}^{N}$ of K parallel-connected capacitors are computed by K times as Equation (1).
(1)$$C_{parallel}^{P,N}=\sum_{k}C_{k}^{P,N}=KC$$
where we assume that the capacitance of single capacitor $C_{k}^{P,N}$ is equal to C so that $C_{k}^{P,N}=C,\;k=1,\;2,\;\cdots ,\;K$. The voltage charged to the K capacitors connected in parallel is expressed as a function of time as follows:(2)$$V_{charge}^{P,\;N}\left(t\right)=V_{S}\left(1-e^{-\frac{t}{Z_{0}KC}}\right)$$
where $Z_{0}$ is the internal impedance of the PVDF energy harvester. The electric charges charged in $C_{k}^{P}$ and $C_{k}^{N}$ are transferred to $C_{store}$ by crossing two switches in the storing stage. At the same time, the switches located in the voltage boosting stage operate alternately according to the current direction. While $C_{k}^{P}$ and $C_{k}^{N}$ are charged, the switches are open and the K capacitors are connected in parallel. On the other hand, while $C_{k}^{P}$ and $C_{k}^{N}$ are discharged, the switches are closed and the K capacitors are connected in series. The total capacitances of $C_{series}^{P}$ and $C_{series}^{N}$ of K series-connected capacitors are computed as follows:(3)$$C_{series}^{P,N}={\left(\sum_{k}C_{k}^{P,N}\right)}^{-1}=\frac{C}{K}$$

The discharged voltage of the K capacitors connected in series is expressed as a function of times, as follows:(4)$$V_{discharge}^{P,N}\left(t\right)=KV_{S}\times e^{-\frac{t}{Z_{L}C/K}}$$
where $Z_{L}$ is the load impedance of $C_{store}$. Therefore, the voltage is boosted K times while $C_{k}^{P}$ and $C_{k}^{N}$ are discharged, and the power can be transferred to $C_{store}$ more quickly.

$C_{store}$ in IC-VBC has a large capacitance to store a lot of energy. Therefore, $Z_{L}$ is computed as the inverse function of $C_{store}$ and has a low value. On the other hand, the PVDF energy harvester has a high internal impedance with a thin thickness. This impedance difference causes a large leakage in power transfer. The proposed IC-VBC delivers power to the load by reducing the impedance of the PVDF energy harvester via a cross-switching operation according to the current direction of $V_{s}$. The matched impedance $Z_{M}$ is determined as in Equation (5) by the capacitance and number of capacitors constituting the voltage boosting stage.
(5)$$Z_{M}=\frac{1}{2\pi f\frac{C}{K}}$$
where f is the fundamental frequency of $V_{s}$. The power stored in $C_{store}$ is supplied to the electrical load $Z_{A}$ to drive applications such as sensors and electronic devices. The cross-switching function of the IC-VBC is implemented using MOSFET transistors. Figure 2 shows the schematic circuit diagram of IC-VBC with K=3.

Figure 3 shows the working flow chart of IC-VBC. The working flow of IC-VBC is as follows:
(1)At positive $V_{s}$, K capacitors of $C_{k}^{P}$ are connected in parallel and the charged voltage is expressed as Equation (2).(2)At negative $V_{s}$, the switches are closed and the K capacitors are connected in series. These series connection increasing voltage K times delivers the power to $C_{store}$. The charged voltage is expressed as Equation (4). Note that the switches are opened and the K capacitors $C_{k}^{N}$ connected in parallel are charged with negative $V_{s}$ at the same time.(3)When $V_{s}$ becomes positive again, the $C_{k}^{P}$ are connected in parallel and charged, and the $C_{k}^{N}$ are connected in series to deliver power to the $C_{store}$.


## 3. Simulation Results

For SPICE simulation, we measure the generated voltage of the PVDF energy harvester according to wind power and estimate the source voltage. We measured the generated signals according to wind speeds of approximately 4 m/s, 6 m/s and 8 m/s using a PVDF energy harvester of 155.7 mm × 18.0 mm × 0.157 mm (height × width × thickness). Figure 4 shows the generated signal and frequency spectrum of the PVDF energy harvester measured at 10 MΩ with an oscilloscope (DSO7052B, Agilent Technologies, Santa Clara, CA, USA). The PVDF generates unstable velocity distribution by vortex flow and increases directional deformation proportional to fluid velocity [22]. Therefore the PVDF energy harvester generates a high voltage proportional to the wind speed and includes a fundamental frequency of 2.9 Hz to 3.7 Hz and frequency components of 12 Hz and 35 Hz.

The PVDF energy harvester is equivalent to a circuit model of internal capacitance $C_{0}$ and source voltage $V_{0}$, connected in series [22]. The determined internal capacitance using an LCR meter, $C_{0}$, is found to be 10.26 nF. The $C_{0}$ is similar to the nominal capacitance of 11 nF of the used PVDF (LDT1-028K, TE Connectivity, Schaffhausen, Switzerland). The source voltage, $V_{0}$, can be estimated using a first-order Butterworth inverse filter at the measured voltage at 10 MΩ [22]. Figure 5 shows the estimated source using the measured voltage at 8 m/s wind speed. In Figure 5b, the fundamental frequency of 3.7 Hz has a large difference of more than 10 dB compared to the frequencies of 11 Hz and 35 Hz. The internal impedance of the PVDF energy harvester is calculated and found to be $1/2\pi fC_{0}=4.19\;M\Omega$ with f=3.7 Hz.

Conventional bridge diode rectifier circuits, voltage doubler rectifiers, and charge pump rectifier circuits are used for energy harvesting [22,32,33]. Half wave rectifier, bridge diode rectifier, voltage doubler, and charge pump rectifier circuits were considered as shown in Figure 6 and their average powers according to load were compared with the IC-VBC via SPICE simulation. For simulation, we set K=2, C=170 nF so that $Z_{m}=500\;k\Omega$ in Equation (5) and implemented the IC-VBC using Infineon’s BSP92P and IRL510 MOSFET transistors and Vishay’s BAT43 diode. The comparison results are shown in Figure 7 by using the same Vishay’s BAT43 diode for all rectifier circuits.

Figure 7a shows the results obtained by using a source voltage of 3.7 Hz and assuming constant directional deformation of PVDF over time. The proposed IC-VBC can match the 500 kΩ impedance of the PVDF and supply 1.6 times higher power than the bridge diode rectifier circuit. Figure 7b showing the simulation with source voltage estimated from measured data, shows that the IC-VBC reduces the impedance of the PVDF from 4.19 MΩ to 320 kΩ and supplies 1.52 times higher power than the bridge diode rectifier circuit.

The voltage multiplier circuits such as the voltage doubler rectifier and the charge pump rectifier use two capacitors in series for voltage boosting. The connected capacitors inevitably increase the impedance of the PVDF energy harvester and cause power transfer loss. Therefore, the voltage multiplier circuits supply lower power than the IC-VBC and the bridge diode rectifier circuit.

We set $C_{store}=10\;\mu F$ for various applications as described in Appendix B and compared the charged voltage according to the number of capacitors in the voltage boosting stage and wind speed. As described in Appendix A, C is set differently depending on the number of capacitors so that $Z_{m}=500\;k\Omega$ in Equation (5). Figure 8a shows the charged voltage in $C_{store}$ according to the number of capacitors in the voltage-boosting stage. The charged voltage is 1.49 V when a single capacitor is used in the voltage boosting stage. The voltage is boosted 1.95 times to charge 2.90 V when two capacitors are used and boosted 2.52 times to charge 3.75 V when three capacitors are used. Conventionally, passive devices can have a characteristic error within ±15% [34,35]. The characteristic error of the internal resistance of the diodes and capacitors in the IC-VBC causes a deviation of the voltage charged in $C_{store}$. Therefore, the voltage charged in $C_{store}$ may have a deviation of 1.12 V due to the characteristic error of the passive devices when three capacitors are used. Figure 8b shows the voltage charged in $C_{store}$ according to the wind speed. Three capacitors were used in the voltage-boosting stage of the IC-VBC. The charged voltage is 1.98 V when the wind speed is 4 ms, and we can charge approximately 0.9 V elevated voltage when the wind speed increases by 2 m/s.

## 4. Experiment Results

We implemented IC-VBC connected to the PVDF energy harvester and verified the integrated system through experiments as sown in Figure 9. The PVDF energy harvester and the IC-VBC was implemented identically to the SPICE simulation. Figure 10 shows the charged into $C_{store}$, which is obtained by an average of 10 experiments by changing the number of capacitors and PVDFs. Figure 10a shows the voltage charged in $C_{store}$ according to the number of capacitors at a wind speed of approximately 8 m/s. The charged voltages are 1.43 V and 2.68 V when a single and two capacitors are used in the voltage boosting stage. The charged voltage is 3.38 V, which is 2.36 times higher than when a single capacitor is used, or when three capacitors are used. These results are within the range of deviation due to a characteristic error of ±15% in the SPICE simulation. The voltage charged in $C_{store}$ reaches the maximum voltage after approximately 600 s regardless of the number of capacitors in the voltage boosting stage.

Multiple PVDFs were attached as shown in Figure 9a and connected in series to the IC-VBC. Figure 10b shows the voltage charged in $C_{store}$ according to the number of PVDFs. We used three capacitors in the voltage boosting stage of IC-VBC and measured the voltage at a wind speed of approximately 8 m/s. Subsequently, we can charge a voltage of 3.73 V using three PVDF energy harvesters.

For a practical application of the suggested IC-VBC, we demonstrated the charging of a conventional lithium-ion battery. We charged Seiko’s MS621FE lithium-ion battery using the IC-VBC which includes the voltage boosting stage with three capacitors and three PVDFs generating a voltage of 3.73 V as shown in Figure 10b. The MS621FE lithium-ion battery is charged using a voltage of 2.85 V to 3.8 V [36]. We use Analog’s ADP5304 switch to transfer power from IC-VBC to a lithium-ion battery [37]. Figure 11a shows the output voltage of the IC-VBC and the current transferred to the lithium-ion battery. A power of 304 μW is transferred to the lithium-ion battery when the output voltage of the IC-VBC reaches 3.8 V. The battery charging system implemented with the PVDF energy harvester and the IC-VBC has a power transfer cycle of approximately 58 s.

As described in Section 2, the voltage charged in $C_{store}$ increases by 0.9 V as the wind speed increases by 2 m/s. Therefore, the output voltage of IC-VBC, which uses three PVDFs and three capacitors in the voltage boosting stage, according to the wind speed is estimated as Equation (6) based on the output voltage of 3.73 V at 8 m/s wind speed as described in Appendix C.
(6)$$V_{out}\left(s\right)=3.73-0.45\left(8-s\right)$$
where s is wind speed. Figure 11b shows the output voltage of IC-VBC according to the wind speed and the applications that can be utilized. The lithium-ion battery can be charged at a wind speed of approximately 5.9 m/s or higher, as shown in the previous experimental results. We can generate a 1.8 V driving voltage for a wireless communication module for IoT applications at wind speeds over 3.7 m/s [38]. In addition, we can drive temperature and humidity determination sensors capable of low voltage operation with low-speed wind power generation of 3.7 m/s or less [39,40].

## 5. Conclusions

We propose the IC-VBC for efficient power transfer of the PVDF energy harvester with high internal impedance and low voltage output. The proposed IC-VBC consists of the voltage boosting stage with multiple capacitors and the storing stage that delivers power to the electrical load. Capacitors of the voltage boosting stage are connected in parallel to the charge, and are connected in series via MOSFET switching to transfer power. Capacitors connected in series transfer power efficiently by boosting the voltage and reducing the total capacitance. The MOSFET transistors in storing stage automatically couple the impedance between the PVDF energy harvester and the electrical load. Therefore, the IC-VBC can reduce the impedance of the PVDF energy harvester during the power transfer process. SPICE simulation results show that IC-VBC reduces the impedance of the PVDF energy harvester from 4.3 MΩ to 320 kΩ and increases the output voltage by 2.52 times. We successfully charged lithium-ion batteries using the PVDF energy harvester and IC-VBC with low-speed wind power generation.

## Figures and Tables

**Figure 1 sensors-23-00137-f001:**
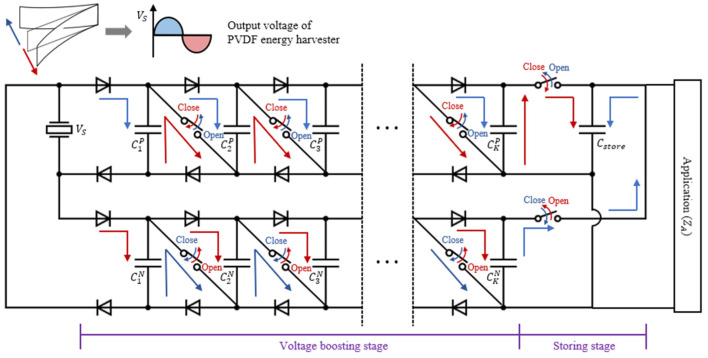
Working principle of IC-VBC.

**Figure 2 sensors-23-00137-f002:**
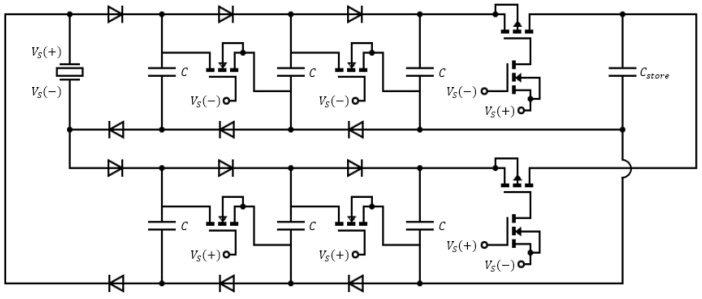
Circuit diagram of IC-VBC (K=3).

**Figure 3 sensors-23-00137-f003:**
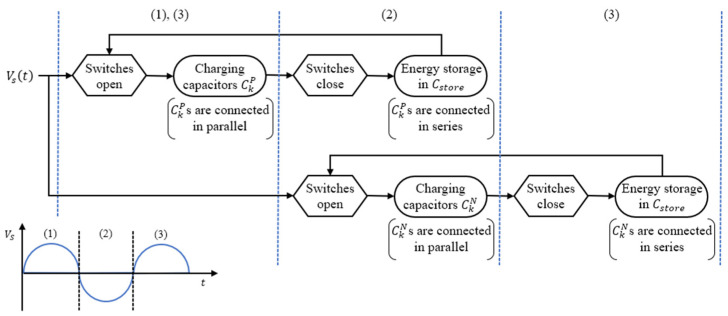
Working flow chart of IC-VBC.

**Figure 4 sensors-23-00137-f004:**
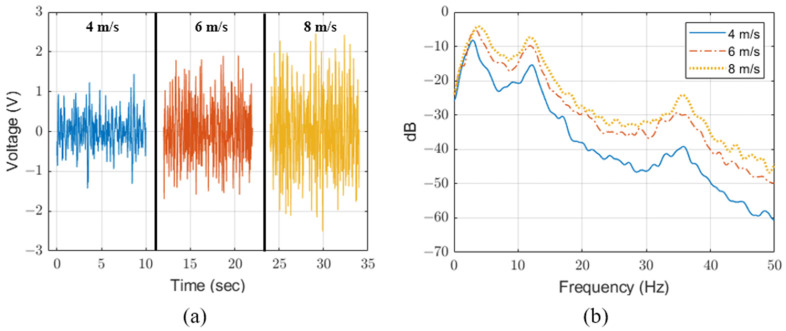
(**a**) Output voltage and (**b**) frequency spectrum of PVDF energy harvester.

**Figure 5 sensors-23-00137-f005:**
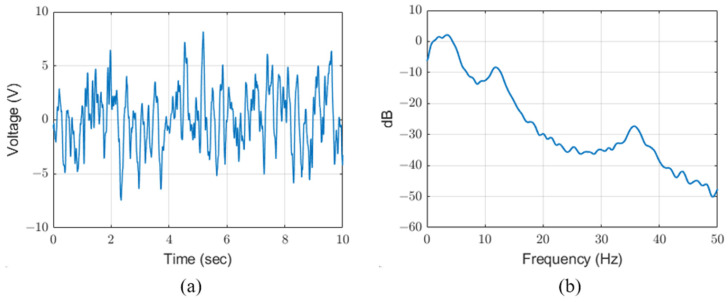
(**a**) Estimated source voltage and (**b**) frequency spectrum of PVDF energy harvester at 8 m/s wind speed.

**Figure 6 sensors-23-00137-f006:**
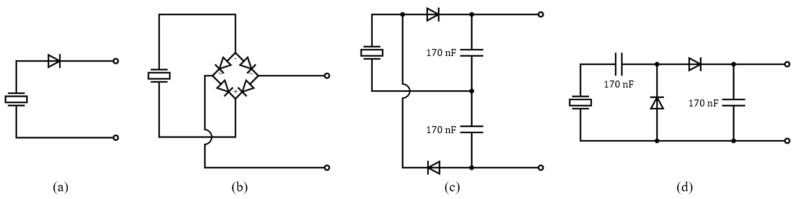
(**a**) Half wave rectifier, (**b**) bridge diode rectifier, (**c**) voltage doubler rectifier and (**d**) charge pump rectifier circuit.

**Figure 7 sensors-23-00137-f007:**
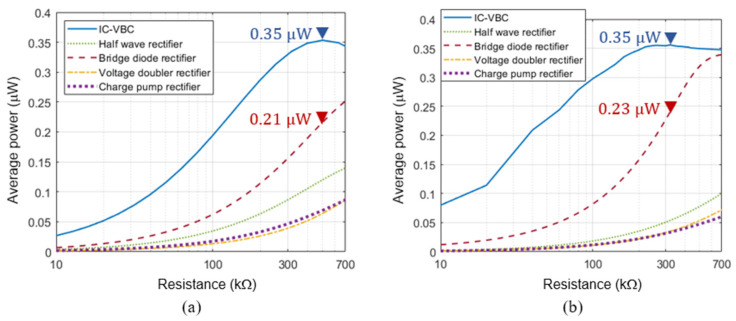
Comparison of average power between IC-VBC and conventional rectifier circuits using (**a**) simulated source voltage and (**b**) estimated source voltage from experimental data.

**Figure 8 sensors-23-00137-f008:**
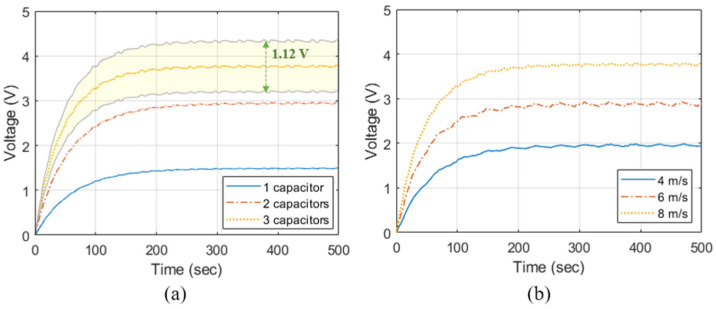
Charged voltage according to (**a**) number of capacitors at 8 m/s wind speed and (**b**) wind speed with 3 capacitors.

**Figure 9 sensors-23-00137-f009:**
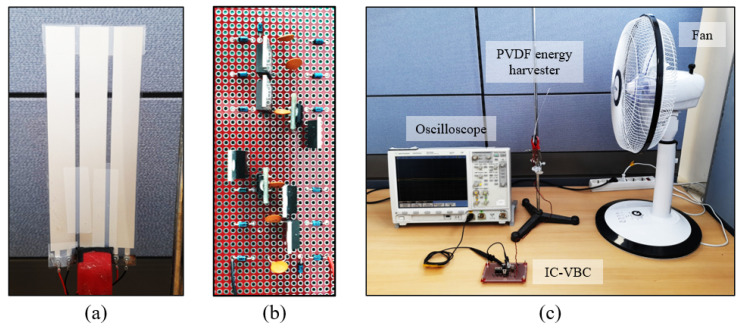
(**a**) PVDF energy harvester, (**b**) IC-VBC, (**c**) experimental setup.

**Figure 10 sensors-23-00137-f010:**
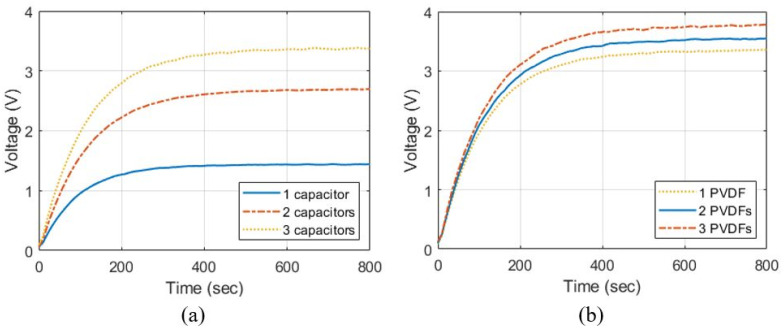
Charged voltage according to (**a**) number of capacitors with single PVDF at 8 m/s wind speed and (**b**) number of PVDFs with 3 capacitors at 8 m/s wind speed.

**Figure 11 sensors-23-00137-f011:**
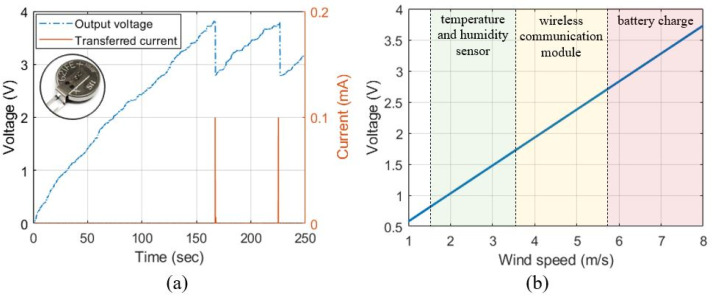
(**a**) Output voltage of IC-VBC and current transferred to lithium-ion battery, (**b**) estimated output voltage of IC-VBC according to wind speed.

## Data Availability

Not applicable.

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
