# Peer review of "Impedance Coupled Voltage Boosting Circuit for Polyvinylidene Fluoride Based Energy Harvester"

_sensors, 2022, doi:10.3390/s23010137_

Round 1

Reviewer 1 Report

There are some issues needed to be addressed before giving out the final judgement. 1. The novelty of the paper is not clear. There is some related work which the authors should clarify what is the main differences with these work. The authors should revise the introduction section significantly

2. The authors should add some clear explanations and a flowchart about the proposed solution. The novelty of the proposed solution is not clear.

3. Simulation results are very incomplete. Please use more schemes to compare your work.

4. There are some typos and grammatical errors. The paper should be written and checked very carefully.

Reviewer 2 Report

In order to reduce the internal resistance of PVDF type energy harvester and improve its output voltage, an IC-VBC circuit is designed in this paper, and simulation analysis and experimental research are carried out, but the work is not sufficient. It is recommended to be published in other journals. The following contents of the paper need to be improved and modified.

1.      The paper is not sufficient for domestic and foreign literature research, and it is suggested to supplement

2.      As shown in Fig. 3a and Fig. 4a, the voltage waveform obtained from simulation analysis is unstable. Please analyze the cause. Figure 3a and Figure 4a, why is the difference of voltage amplitude large when the wind speed is 8m/s? How is 10MΩ obtained during simulation analysis?

3.      line123-125:The proposed IC-VBC reduces the impedance of the PVDF energy harvester from 4.19MΩ to 320kΩ。 How to realize the system internal resistance from 4.19 MW to 320?

4.      line129:“The PVDF energy harvester was matched to 320kΩ with an error of 180kΩ compared to Zm=500kΩ”ï¼Œ What is the cause of the error?

5.      figure6:The internal resistance of the three circuits is different due to their different compositions. During simulation, the matching resistance is 320kΩ. Is it reasonable? According to Figure 6, when the IC-VBC circuit is used for experiment, the effective power has become stable when the matching resistance is 230 kΩ. Why does the paper take the matching resistance as 320 kΩ?

6.      Figure7 : The simulation experiment analyzes the influence of the number of capacitors on the output voltage of the energy harvester. What is the reason for the increase of the number of capacitors and the increase of the output voltage? Is it caused by the small capacitance of the capacitor selected in the paper? If it has nothing to do with the capacitance value of the capacitor, how many capacitors are designed in this experiment for a relatively optimal choice?

7.      Line138: In the article "Conventionally, passive devices can have a characteristic error within ± 15%", references should be given.

8.      Figure 8: The figure shows that the wind source is the fan during the experiment, and the fan outputs the periodic unstable wind, which will cause the instability of the experimental results. It is recommended to select the wind tunnel with stable output as the wind source.

9.      Figure9 andfigure7:Figure 7 shows that when there are 1, 2 and 3 capacitors in the circuit during simulation analysis, the output voltages are 1.49V, 2.90V and 3.75V respectively. Figure 9 shows that when there are 1, 2 and 3 capacitors in the circuit during experimental analysis, the output voltages are 1.22V, 2.39V and 3.32V respectively. Line156-157: "These results are within the range of development due to a characteristic error of ± 15% in the SPICE simulation".According to the data in the paper, when the capacitor is 1 or 2, the error between the experimental value and the simulation value is greater than 15%, which is inconsistent with the explanation of the author of the paper. Please explain the cause.

10.   It is suggested to supplement the derivation process of formula (6).

11.   The paper lacks experimental data comparison with similar circuits, and it is suggested to supplement

Reviewer 3 Report

The author propose a way to reduce the impedance of PVDF energy harvester while incresing the voltage. As far as I could understand the idea consists in accumulating energy in a set of capaitors connected initially in  parallel during the charging process and later in serie during discharging. Theory, simulation and experimental stages are well described, showing that impedance can be reduced  while voltage would be increased. However my main issue is how much novelty this proposal contains. The reader of this journal would like to ask where is the new physical insight that this neat and respectable article  provides. 

Round 2

Reviewer 2 Report

The author has improved and revised the paper according to the review comments. I have no other suggestions.

Reviewer 3 Report

The author have given appropriate response to my concerns and I can now understand the relevance and originality of the proposal. I recommend publication of the article.